# Daily replacement of very high-fat diet stabilizes food intake and improves mouse welfare by ensuring food quality

Joshua Cordeira [ORCID] *

Department of Biology, Western Connecticut State University, Danbury, Connecticut, United States of America

* CordeiraJ@wcsu.edu

**Data Availability Statement:** Data underlying results presented in the study are available at http://doi.org/10.6084/m9.figshare.23662566

**Funding:** This work was supported by Grants for Laboratory Animal Science (GLAS) from the

## Abstract

Researchers are obligated to ensure food quality and provide laboratory animals with a palatable diet. Factors influencing the quality and palatability of very high-fat diet (VHFD), a widely used rodent diet, however, are understudied. We conducted experiments to establish best practices for ensuring the quality of VHFD and to improve mouse welfare. We found that VHFD in the food hopper was vulnerable first to dehydration and then oxidation within 7-days, leading to dramatic changes in food intake and food preference behavior in mice. Mitigating dehydration and oxidation of VHFD by replacing food daily, rather than weekly, stabilized feeding behavior without effect on overall cardio-metabolic health. Importantly, daily replacement of VHFD also reduced measures of anxiety-like behavior in the open field test. Refining husbandry practices to include daily replacement of VHFD can therefore ensure VHFD quality and improve animal welfare. Standardizing the practice of daily VHFD replacement may also prevent experimental confound and improve experimental reproducibility and replicability.

## Introduction

Providing laboratory mice with high-quality food is a cornerstone of good husbandry practice and is essential to animal welfare. The *Guide for the Care and Use of Laboratory Animals* specifically dictates that "animals should be fed palatable, uncontaminated diets that meet their nutritional and behavioral needs." Because food may become nutritionally deficient, the *Guide* also cautions researchers to implement procedures and practices (e.g., storage and handling) which ensure food quality, especially for perishable foods like high-fat diets [1].

One of the most popular high-fat diets used by researchers is D12492, a very high-fat diet (VHFD) composed of 60% kcal from fat (35% w/w). Procedures and practices which ensure the quality of VHFD, however, are poorly defined. Research Diets, the manufacturer for VHFD, simply recommends to "store frozen." The literature offers little additional guidance. Factors influencing the quality of VHFD are understudied in general. And, despite widespread use of VHFD, research reports rarely include methodological details describing the storage

American Association for Laboratory Animal Science to JC. Additional support was provided by the Connecticut State Universities American Association of University Professors and the Department of Biology at Western Connecticut State University. The funders had no role in study design, data collection and analysis, decision to publish, or preparation of the manuscript.

**Competing interests:** The authors have declared that no competing interests exist.

and handling practices of VHFD. In most cases, frequency of diet change is probably once per week along with routine cage change, but husbandry practices may vary between animal facilities.

Storage and handling practices of VHFD are important because they can impact experimental outcomes. The temperature at which VHFD is stored, for example, differentially alters weight gain and liver damage in mice [2]. Storage conditions of VHFD also affect pulmonary inflammation in mice [3]. Thus, the storage and handling practices of VHFD are important variables to consider, control, and report for researchers to avoid experimental confound and prevent problems with replicability and reproducibility.

How storage and handling conditions of VHFD impact experimental outcome could be related to food quality. A few researchers have speculated that the quality of VHFD may be reduced from spoilage by lipid oxidation and that this can be avoided by replacing food weekly [4] or 2–3 times weekly [5, 6]. But, there is no evidence that such practices ensure the quality of VHFD. In fact, a timeline for the oxidation of VHFD or even the extent to which oxidation occurs, has not been thoroughly investigated. The food replacement schedule which best ensures the quality of VHFD is unknown.

To help establish best practices for ensuring the quality of VHFD and improving animal welfare, in this study, we investigated factors influencing the quality of VHFD and the impacts on mouse feeding behavior and health. Specifically, we measured dehydration and oxidation of VHFD over time and investigated their roles in food intake and food preference behaviors. We also assessed the impact of daily, rather than weekly, replacement of VHFD on measures of physical and mental health in mice. By offering specific recommendations for ensuring food quality and improving animal welfare, results are of practical significance to investigators using VHFD.

## Materials and methods

### Animals

Male C57BL/6NCrl mice were purchased from Charles River Laboratories (Wilmington, MA) at 6 weeks old. Upon arrival, mice were group housed and habituated for 2 weeks. Mice were then individually housed for the duration of experiments. Mice were housed in standard, type-III filter-topped cages (425 x 276 x 153 mm, floor area 820 cm$^2$, Tecniplast, West Chester, PA) with cob bedding (1/4-in Bed-o'Cobs, Andersons Inc., Maumee, OH) and nestlet (Ancare Corp., Bellmore, NY). Mice were maintained on a 12h-12h light-dark cycle in a climate-controlled room with a temperature of 24–26˚C and 20–60% relative humidity. Mice had unlimited access to water and food. All experimental procedures complied with recommendations in *the Guide for the Care and Use of Laboratory Animals* of the National Institutes of Health [1] and were approved by the Institutional Animal Care and Use Committee at Western Connecticut State University (protocol 2022–05).

### Food

All mice had unlimited access to standard rodent diet (#5001; LabDiet, St. Louis, MO) during the 2-week habituation period upon arrival. Afterward, mice were provided with very high-fat diet (VHFD; D12492, Research Diets Inc., New Brunswick, NJ). Access to VHFD depended on the experiment, described in detail below. VHFD is composed of 60% kcal (35% w/w) from fat. The formulation of VHFD is presented in S1 Table.

The manufacturer ships VHFD without ice and recommends that it be stored frozen. So, we aliquoted and froze VHFD immediately upon arrival. We aliquoted VHFD in Ziplock® bags and stored aliquots within the sealed, opaque packaging provided by the manufacturer.

VHFD was frozen at -20˚C. To avoid multiple freeze/thaw cycles of the lot, aliquots were removed from the freezer as needed. VHFD was thawed at room temperature for 1 h before being provided to mice. Testing using an infrared surface thermometer revealed that 1 h was sufficient for frozen VHFD to reach room temperature (S1 Fig).

## Experimental design

We used several experiments to investigate interactions between mouse behavior and the quality of VHFD. Methodological details are included in the sections that follow. In brief, we first measured daily food intake to study the pattern of VHFD intake on different days of the week. We then performed food preference tests to assess changes in the palatability of VHFD over time. To evaluate factors influencing the quality of VHFD, we measured food dehydration and oxidation over time. Next, we directly manipulated the dehydration and oxidation status of VHFD to measure impacts on food intake and preference. We finally tested whether using daily food replacement to control food quality had effects on intake behavior, anxiety-like behavior, and cardiometabolic health in mice.

## Daily food intake

We used daily food intake to begin to study the pattern of VHFD intake on different days of the week. Mice (n = 6) were maintained on VHFD, available *ad libitum*, for 4 weeks. Daily food intake was calculated from the change in the mass of VHFD, measured daily. VHFD was replaced in the feed hopper once per week on the same day (Monday) each week. All pieces of food remaining in the feed hopper or on the floor of the cage were weighed, discarded, and replaced by freshly thawed VHFD. At the start of week 5, VHFD was replaced on Tuesday rather than Monday to assess "weekend effects." Cages were changed on a separate day (Saturday) each week to control the potentially confounding influence of cage change on food intake.

## Food preference

We used food preference testing to assess the palatability of VHFD. Food preference testing was performed for 2 h, beginning at light onset on Monday, Wednesday, and Friday of each week. Mice (n = 8) were non-food deprived and maintained on an *ad libitum* standard rodent diet. During the test, standard rodent diet was removed, and mice instead ate from two sources of VHFD separated by a metal divider in the feed hopper. Mice could choose to eat "fresh" VHFD which had been removed from the freezer and allowed to thaw to room temperature or "old" VHFD which had been thawed and left in the feed hopper of an empty cage for 1, 2, 3, 4, 5, 6 or 7-days. Food intake was calculated from the change in food mass during the food preference test. Food preference (%) was calculated using the formula: [% = (fresh or old VHFD intake / total VHFD intake) x 100]. Mice were given similar amounts of food on both sides of the food hopper. To eliminate side bias, mice performed each preference test twice, with foods assigned to alternating sides of the feed hopper, and the results averaged. To avoid order effects, food preference tests were conducted in pseudorandom order. Mice performed the food preference using fresh VHFD on both sides of the hopper test 3 times before experimentation, first to eliminate novelty and twice for control.

## Dopamine

We also assessed palatability of VHFD by measuring dopamine release after consumption. Mice (n = 8) were maintained on standard diet and given access to freshly thawed VHFD or

7-day-old VHFD (alternating days) for 30 minutes at light onset. Mice were euthanized 15 minutes after consuming VHFD on day 7. Brains were quickly extracted and placed into ice cold 1x PBS. The Nucleus Accumbens (core and shell) was bilaterally dissected using a 1.5 mm biopsy punch from 1 mm coronal sections. Tissue samples were snap frozen and shipped to the Neurochemistry core at Vanderbilt University for quantification of monoamine neuro-transmitters using HPLC with electrochemical detection. DA release was estimated using the ratio of 3-methoxytyramine (3-MT) to DA [7] and compared between fresh and 7-day-old VHFD conditions (n = 4 each).

## Food quality testing

**Water content.**   To evaluate food moisture, VHFD samples (n = 15) were left in empty cages and samples were weighed after VHFD was thawed (0 d) and again at 1, 2, 3, 4, 5, 6, and 7 days. Note: at this time during testing, the relative humidity of the testing room was 20–30%. Change in mass was calculated as the cumulative percent change in mass from the starting mass. Samples of VHFD (n = 6 each) were also maintained in storage containers at ∼80%, 70%, 60%, 50%, 40%, and 30% relative humidity using humidity control packs (Boveda Inc., Minneapolis, MN) to examine the role of relative humidity on the water content and mass of VHFD over time.

**Non-specific lipid oxidation.**   A Thiobarbituric Acid Reactive Substances (TBARS) assay (Oxford Biomedical Research, Inc., Rochester Hills, MI) was used to measure the formation of lipid peroxides and aldehyde products in VHFD samples. VHFD samples were thawed and left at room temperature for 0–7 days in preparation for testing. The assay was performed accord-ing to manufacturer instructions. TBARS were measured against a standard curve of known Malondialdehyde (MDA) concentrations. All samples were tested on the same day. Samples were measured in triplicate and the results averaged.

**Primary oxidation.**   AOCS method Cd 8b-90 [8] was performed with slight modification to measure peroxide value (POV), the milliequivalents of peroxide (the primary product of fat oxidation) per 1 kg (meq/kg) of fat in VHFD samples. AOCS method Cd 8b-90 is a highly empirical titration. Thus, samples from 0–7 days were timed to be tested together on the same day, using the same protocol, and same set of reagents. Samples were weighed then dissolved in 30 mL of 3:2 v/v acetic acid-isooctane solution. Next, 1 mL of saturated potassium iodide solution was added, and the solution mixed for 10 minutes while shaking. Finally, 30 mL of water was added, and the sample mixture was titrated with 0.01 N sodium thiosulfate using a starch indicator. POV was calculated using the following formula: POV = [(S-B) x N x 1000]/ M, where: B = mL of titrant used in blank, S = mL of titrant in sample, N = normality of titrant, M = g mass of fat sample. VHFD contains 35% fat (w/w) (S1 Table). VHFD without dye (D12492N, Research Diets Inc., New Brunswick, NJ) was used to avoid the possibility that the dye would interfere with end-point color detection. All samples were tested in quadruplicate and the results averaged.

**Secondary oxidation.**   AOCS method Cd 18–90 [9] was used to measure p-Anisidine Value (p-AV) of VHFD samples. The p-AV is a direct measure of aldehydes and secondary oxidation using spectrophotometry. In brief, VHFD samples were added to isooctane and absorbance measured in a quartz cuvette at 350 nm against a pure isooctane blank. Anisidine reagent was added to each of the samples and incubated for 10 minutes in the dark. Absor-bance was measured again at 350 nm, against an isooctane + anisidine blank. The p-AV was calculated using the following formula: $[10 \times (1.2A_s-A_b)]/m$ where $A_s$ = absorbance of fat solu-tion after reaction with anisidine, $A_b$ = absorbance of fat solution, m = mass of fat sample (g). VHFD samples from 0–7 days were timed to be tested together on the same day, using the

same protocol, and same set of reagents. VHFD without dye (D12492N) was again used to minimize interference with spectrophotometry. All samples were tested in quadruplicate and the results averaged.

## Effects of water content and oxidation of food preference and intake

To assess the role of food moisture in VHFD preference, mice (n = 8) were fed standard chow and subjected to food preference tests as described earlier, except that the moisture content of VHFD was directly manipulated. Tests included fresh versus dehydrated VHFD or fresh versus rehydrated 1, 2, 3, 4, 5, 6, or 7-day-old VHFD. To create dehydrated food, pelleted VHFD was crumbled and dried in a vacuum desiccator for 24 h in the dark. Rehydrated VHFD was created by replacing mass lost in 1, 2, 3, 4, 5, 6, or 7-day-old VHFD with 1 mL of water per 1 gram of mass lost. VHFD was re-pelleted manually using a 15 mL tube and a pestle. Fresh VHFD was treated similarly (thawed, crumbled and re-pelleted) for control.

To assess the roles of food moisture and oxidation in weekly VHFD hyperphagia, a separate group of mice (n = 8) were fed VHFD *ad libitum* and food intake was calculated, as before. During weekly food replacement, existing VHFD in the feed hopper was replaced with either: freshly thawed VHFD, 7-day old VHFD (to control for the act of replacing food), dehydrated VHFD, or rehydrated 7-day old VHFD.

## Effects of food replacement frequency on anxiety-like behavior and cardiometabolic health

Mice were randomly divided into two groups (n = 12 each) and maintained on VHFD, available *ad libitum*, for 8 weeks. For one group of mice (Weekly mice), VHFD was replaced once per week on the same day (Monday) each week. For the other group of mice (Daily mice), VHFD was replaced daily. Food intake was calculated from the change in the mass of VHFD, measured daily. Body mass was measured weekly, at cage change. Cages were changed once per week on the same day (Friday) each week. After 8 weeks, anxiety-like behavior and cardiometabolic health were measured.

Mice were tested for anxiety-like behavior using the open field test. The test was performed at light onset. Each mouse was placed into a brightly lit (300 lux) open field arena (L x W x H: 40 cm x 40 cm x 30 cm, Stoelting Co., Wood Dale, IL) composed of an opaque gray acrylic plastic. Mice explored the open field for 5 minutes while being video recorded. Any-Maze (Stoelting Co., Wood Dale, IL) software tracked the mouse, measuring time spent in the inner and outer zones. The inner zone was defined as the inner 20 cm x 20 cm. Time spent in the inner and outer zones are validated measures of anxiolytic and anxiety-like behavior, respectively [10]. Fecal pellets deposited during the test session were counted as an additional measure of anxiety and emotionality during the test [10]. The open field was cleaned with 95% ethanol between animals to eliminate olfactory cues. Lastly, fecal pellets deposited during 6 h after a cage change were counted as an additional measure of stress and anxiety.

Total food intake, final body weight, feed efficiency, adiposity and blood levels of glucose, cholesterol, and triglycerides were measured after 8 weeks of weekly or daily VHFD replacement to assess the impact of such treatment on cardiometabolic health. Total food intake and final body weight were used to calculate feed efficiency (grams of weight gain per gram of VHFD consumed). Total adiposity was calculated by dissecting then totaling the wet mass of gonadal, peri-renal, and mesenteric white adipose tissue. Blood glucose levels were measured using a True Metrix blood glucose meter (Trividia Health Inc., Fort Lauderdale, FL) from a drop of tail blood taken after a 6-hour fast. Measurements were taken in triplicate and the results averaged. Cardiac puncture provided blood samples to measure serum lipids. Blood

was allowed to coagulate for 2 h at room temperature before centrifugation for 20 min at 2000 x g. Total serum cholesterol and triglycerides were measured using a Cardiochek PA Analyzer (PTS Diagnostics, Whitestown, IN) with lipid panel test strips (PTS Diagnostics, Whitestown, IN). Mice were euthanized via carbon dioxide inhalation.

## Statistical analysis

Statistical analysis was performed using Jamovi (Sydney, Australia) software [11]. All tests are indicated in Results & discussion. Repeated Measures ANOVA was used to assess changes in daily food intake, food preference, or food quality over time. When main effects were significant, Tukey comparisons were used for post hoc analysis. Paired sample T-Tests were used for planned comparison between time points. One Sample T-Tests were used to determine whether food preference differed significantly from 50% chance. Independent samples T-Tests were used to compare means between Daily and Weekly mice. Mann-Whitney U-Tests were used when data were found to be not normal (Shapiro-Wilk). In the case of unequal variance between groups (Levene's), Welch T-Test was used. For all tests, a p value $< 0.05$ was considered statistically significant.

## Results and discussion

### VHFD intake varied across days of the week

We previously noticed that mice fed very high-fat diet (VHFD) ate more food at the beginning of each week ([unpublished]). To confirm the unpublished observation, we fed mice VHFD *ad libitum* and measured daily intake for 4 weeks. Daily intake peaked each Monday then declined until the next week (Fig 1A). Day of the week significantly impacted intake [RM ANOVA: $F_{(6,30)} = 20.4$, $p < 0.001$, $\eta p2 = 0.803$] such that intake on Monday was significantly greater than any other day of the week [Tukey Post Hoc Comparisons, all $p < 0.05$] (Fig 1B). These results show that mice were hyperphagic on Mondays.

### Weekly VHFD replacement caused hyperphagia

Next, we sought to understand why food intake was elevated each Monday. Since hyperphagia always occurred on the day we replaced food in the feed hopper (Monday), we reasoned that food replacement could be the cause. To test this, at the beginning of week 5, we replaced VHFD on Tuesday instead of Monday. This also helped to rule out whether hyperphagia was due to "weekend effects" (differences in building and/or personnel activity during the weekend versus the workweek).

In support of our hypothesis, food intake increased on Tuesday when food was replaced, rather than on Monday when the workweek began (Fig 1A, note the arrow at the beginning of week 5). Because food replacement and cage change occurred on separate days (Monday versus Saturday, respectively), we also ruled out the possibility that cage change caused food intake to increase. In fact, cage change appeared to reduce food intake [Paired Samples T-test: $t(5) = 2.28$, $p = 0.072$, $d = 0.931$] when compared to the day before (Fig 1B). These results suggest that hyperphagia was due to food replacement, not from weekend effects or cage change.

### Palatability of VHFD declined as it aged

Next, we investigated why food replacement caused hyperphagia. We hypothesized that the palatability of VHFD could change over time, influencing intake. VHFD is highly palatable but "old" VHFD left in a cage for up to 7-days may be less palatable than "fresh" VHFD removed from the freezer and thawed for 1-hour (to reach room temperature). We tested our hypothesis

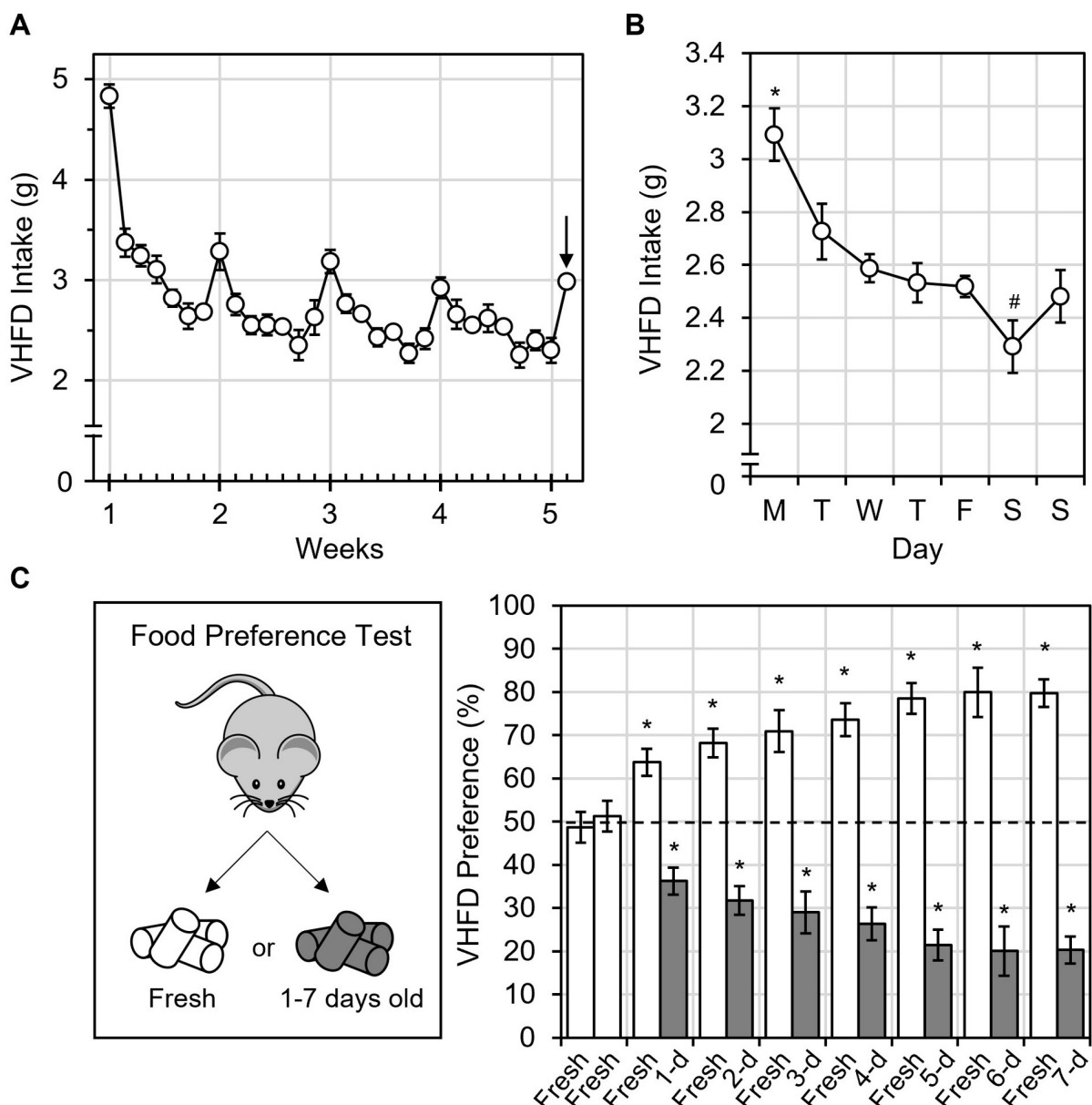

**Fig 1. VHFD intake and palatability are not stable during the week.** (A) Daily intake peaked every Monday, when VHFD was replaced in the feed hopper, then gradually declined until VHFD was replaced again. At week 5, VHFD was replaced on Tuesday (arrow) rather than Monday to rule out "weekend effects." (B) Average intake per day of the week was highest on Monday. (C) Food preference tests show that mice preferred to eat freshly thawed VHFD rather than up to 7-day old VHFD. The dotted line indicates 50% chance. All values represent AVG ± SEM. #p < 0.10, **p < 0.05, *p < 0.01, d = day.

by conducting food preference tests, during which we simultaneously offered mice "fresh" and "old" VHFD left out for 1, 2, 3, 4, 5, 6 or 7 days. We measured the intake of fresh and old VHFD during the 2-h food preference test to calculate food preference/avoidance behavior.

Mice exhibited chance levels of preference (50%) only when fresh VHFD was paired with fresh VHFD, as a control (Fig 1C). Otherwise, mice always preferred to eat freshly thawed VHFD and avoided eating old VHFD which had been left out for 1, 2, 3, 4, 5, 6, or 7 days (Fig 1C). One sample T-tests indicated preferences were significantly different from 50% chance

(p < 0.01) for all food preference tests except fresh vs. fresh. A repeated measures ANOVA revealed a significant effect of time [F(7,49) = 6.93, p < 0.001, ηp2 = 0.497] indicating that preference to eat fresh VHFD and avoid old VHFD increased as the age of VHFD increased. The greatest preference to eat fresh VHFD and avoid old VHFD was when fresh VHFD was tested with 7-day-old VHFD. These results show that the palatability of VHFD significantly declined after it was removed from storage, even after just one day.

As another way to assess palatability, we also measured dopamine release in the nucleus accumbens after mice consumed freshly thawed VHFD or 7-day-old VHFD. Consumption of palatable, high-fat food promotes dopamine release in the nucleus accumbens [12]. Dopamine release was significantly greater for 7-day-old VHFD compared to freshly thawed VHFD [Independent samples T-test: t(6) = -3.98, p = 0.007, d = -2.81] (S2 Fig). This result was surprising since we predicted less dopamine release after consuming older, less palatable VHFD. Interestingly though, Bassareo et al. [12] demonstrated that accumbens dopamine release was elevated by aversive taste stimuli. And, unlike positive stimuli, dopamine released from aversive taste stimuli did not undergo habituation [13]. In this context, our results may indicate that 7-day-old VHFD is aversive to mice.

## VHFD water content altered palatability

We next investigated causes for the reduction in food palatability. Interestingly, we discovered that VHFD left out for food preference testing lost mass over time. VHFD lost 1.51 +/- 0.04% mass over 7-days [RM ANOVA: F(7,98) = 187, p < 0.001 ηp2 = 0.930] (Fig 2A). Since the food was left out by itself and was inaccessible to mice, and because food mass was sensitive to relative humidity (S3 Fig), we attributed the decrease in mass of VHFD to water loss. We therefore investigated whether the water content of VHFD contributed to its palatability.

There was a strong, significant relationship between the change in mass of VHFD (from dehydration) and the preference to consume it [Pearson: r(6) = 0.995, p < 0.001] (Fig 2B). Preference for VHFD decreased as VHFD lost mass. This correlation suggested that the water content of VHFD could contribute to its palatability. To test for causality, we directly manipulated the water content of food and performed food preference tests. Dehydrating freshly thawed VHFD using a vacuum desiccator produced a significant 2.0 +/- 0.1% loss of mass [Paired Samples T-test: t(9) = 21.5, p < 0.001, d = 6.81]. When mice had the choice to consume freshly thawed VHFD or dehydrated VHFD, mice overwhelmingly preferred fresh VHFD and avoided the dehydrated VHFD [One Sample T-test: t(7) = 4.63, p = 0.002, d = 1.64] (Fig 2C). Thus, dehydrating VHFD reduced its palatability.

We also tested the effect of rehydrating VHFD (with 1 mL of water added per 1 gram of mass lost) on food preference. When mice had the choice between fresh VHFD or rehydrated VHFD, results depended on the age of the rehydrated VHFD. Mice consumed rehydrated VHFD with equal preference to freshly thawed VHFD only when rehydrated VHFD was 1, 2, or 3-days old [One Sample T-tests, p > 0.05] (Fig 2D). Mice still avoided eating rehydrated VHFD that was 4, 5, 6 or 7-days old [One Sample T-tests, p < 0.05]. In other words, rehydrating 4, 5, 6, or 7-day old VHFD was not enough to restore palatability. These results indicate that the water content of food is a key factor influencing the palatability of VHFD, but it is not the only factor.

## VHFD oxidation altered palatability

Since VHFD contains 35% (w/w) fat (see S1 Table), we reasoned that lipid oxidation could also decrease VHFD palatability and intake. Others have suggested that oxidation of VHFD

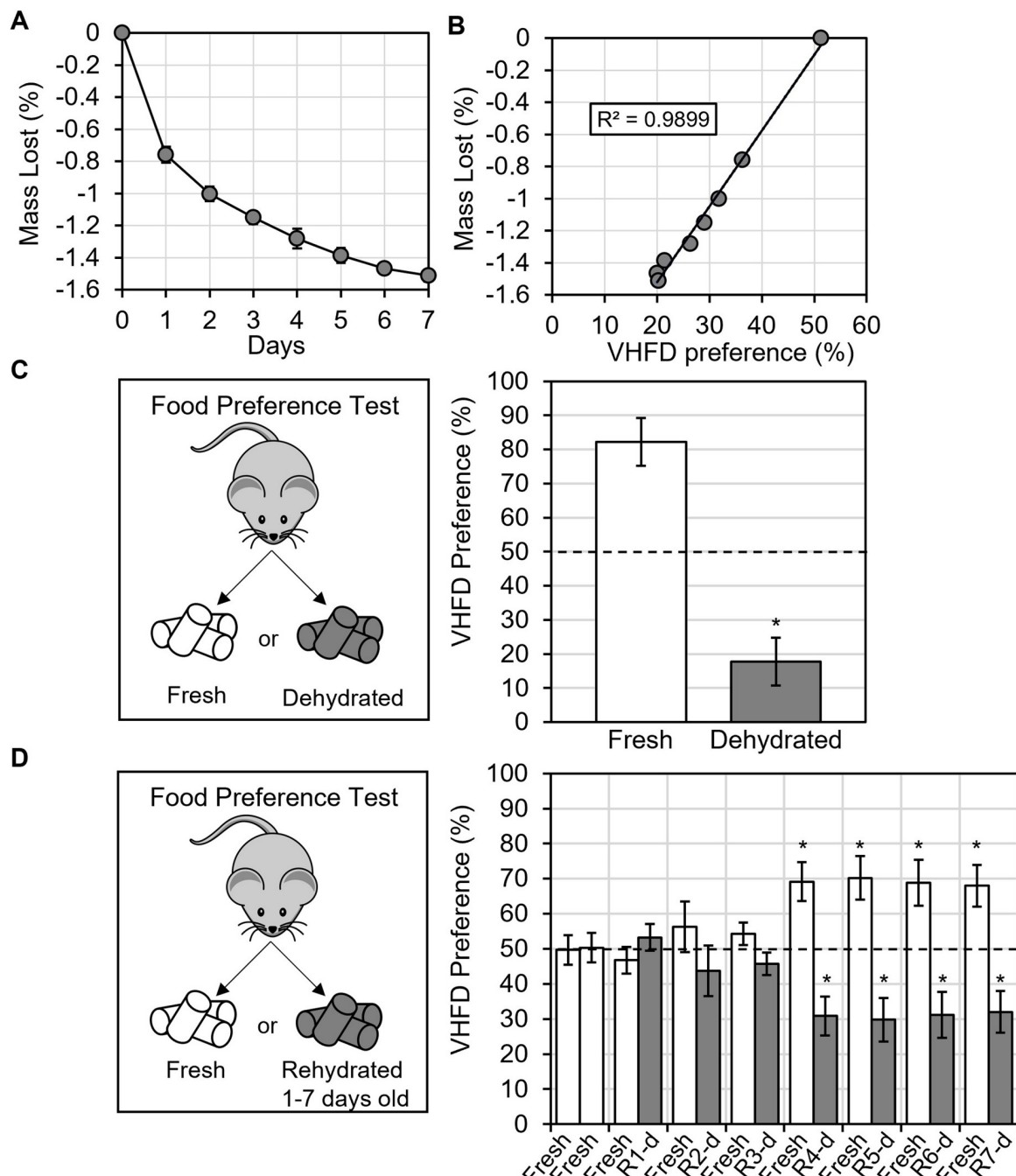

**Fig 2. VHFD becomes dehydrated, decreasing its palatability.** (A) VHFD in an empty cage lost mass over 7-days. (B) Mass lost correlated with VHFD preference. (C) Dehydrating VHFD decreased preference to consume it. (D) Rehydrating VHFD improved preference to consume 1, 2, or 3-day old VHFD but not 4, 5, 6, or 7-day old VHFD. The dotted line indicates 50% chance. All values represent AVG ± SEM. *p < 0.05, *p < 0.01. d = day, R = Rehydrated.

may occur and that it should avoided [5, 6], but this hypothesis has not yet been thoroughly investigated.

We evaluated oxidation in freshly thawed as well as 1, 2, 3, 4, 5, 6, and 7-day old samples of VHFD using multiple techniques. First, we assessed the presence of thiobarbituric acid-

reactive substances (TBARS) as a global measure of lipid hydroperoxides and aldehydes present in VHFD samples [13, 14]. Lipid hydroperoxides produced during primary oxidation are subsequently degraded during a secondary oxidation process which produces volatile aldehydes capable of altering the sensory qualities of food [15]. So, we also assessed lipid hydroperoxide and aldehyde levels individually using AOCS methods [8, 9] to measure peroxide value (POV) and p-Anisidine value (p-AV).

Levels of TBARS in VHFD samples significantly changed over time [RM ANOVA: $F_{(7,14)}$ = 8.94, $p < 0.001$, ηp2 = 0.817], indicating that lipid oxidation occurred (Fig 3A). POV significantly changed over time [RM ANOVA: $F_{(7,21)}$ = 6.5, $p < 0.001$, ηp2 = 0.684], confirming primary oxidation of fat and the presence of hydroperoxides in VHFD samples (Fig 3B). p-AV also significantly changed over time [RM ANOVA: $F_{(7,21)}$ = 3.74, $p = 0.009$, ηp2 = 0.555] confirming secondary oxidation (Fig 3C). These results collectively show that VHFD underwent low levels of lipid oxidation over 7-days (despite inclusion of antioxidant vitamins in the manufacturing process). This is consistent with a recent report that VHFD can become oxidized after a week or more at room temperature [16].

Secondary oxidation most likely explains the changes in food preference during days 4–7, when food preference could not be restored by rehydration. Lipid hydroperoxides produced from primary oxidation are odorless and tasteless [15, 17] and would not be predicted to contribute to changes in VHFD palatability. Only secondary oxidation products are responsible for the undesirable changes in the aroma and flavor properties of foods caused by lipid oxidation [15]. Consistent with this, we observed a peak in primary oxidation at 3 days which declined as peroxides decomposed into secondary compounds during days 4–7 (Fig 3B). Thus, within 7 days, the palatability of VHFD appears to be first reduced from dehydration and later from (secondary) oxidation.

## VHFD oxidation caused hyperphagia from weekly food replacement

Since dehydration and oxidation both contributed to VHFD palatability, we next aimed to determine whether these factors also contributed to hyperphagia following weekly food replacement. We treated mice with VHFD *ad lib* and measured daily intake as before, but deliberately varied the quality of food provided during weekly food replacement.

Food quality significantly impacted overeating after weekly food replacement [RM ANOVA: $(F3,21)$ = 10.1, $p < 0.001$, ηp2 = 0.591] (Fig 3D). Mice again exhibited hyperphagia when 7-day old VHFD was replaced with freshly thawed VHFD (Fig 3D). This was not the case, however, when 7-day old VHFD was replaced with 7-day old VHFD (a control for the act of food replacement). Thus, overeating was not due to the act of food replacement, but rather the replacement of 7-day old VHFD with a more palatable VHFD. We posit that new, freshly thawed VHFD is more palatable by comparison, triggering hyperphagia.

Interestingly, mice did not display hyperphagia when we replaced 7-day old VHFD with rehydrated, 7-day old VHFD (Fig 3D). Since the rehydrated, 7-day old VHFD was still oxidized, this result suggests that oxidation is the more important food quality determining weekly overeating behavior. Consistent with this, we also observed that mice were willing to overeat dehydrated but unoxidized VHFD (Fig 3D). Thus, the oxidation status of VHFD appears to be the primary factor causing hyperphagia following weekly food replacement. Freshly thawed (unoxidized) VHFD may taste or smell better by comparison, leading to hyperphagia.

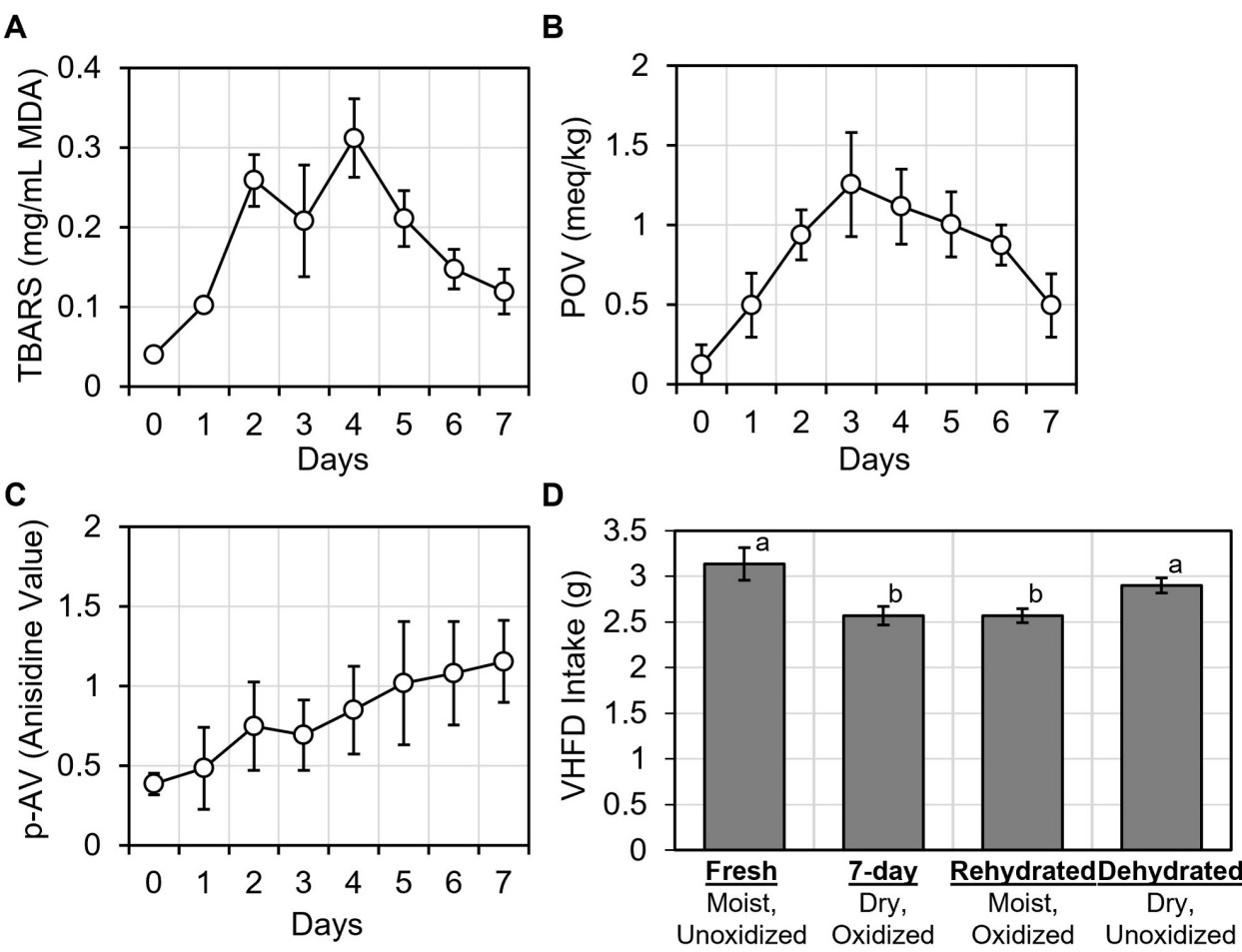

**Fig 3. VHFD becomes oxidized, which alters intake.** (A) Thiobarbituric acid reactive substances (TBARS) in VHFD changed over time. B) Peroxide value (POV), indicative of primary oxidation, peaked after 3-days. C) Anisidine value (p-AV), indicative of secondary oxidation, steadily increased over time. D) Mice only overate unoxidized VHFD after weekly food replacement, regardless of moisture content. All values represent AVG ± SEM. Different letters indicate statistically significant differences between treatments.

### Daily food replacement stabilized VHFD intake

We next investigated whether daily food replacement, which provided mice with the most palatable VHFD every day, could stabilize food intake across the week. We also assessed whether the food replacement schedule affected weight gain and cardio-metabolic health. To test these hypotheses, we fed mice VHFD *ad libitum*, divided mice into two groups, and treated them with either daily food replacement (daily mice) or weekly food replacement (weekly mice) for 8 weeks. We calculated food intake daily and averaged intake by day of the week. At the end of the experiment, we also measured/calculated total intake, final body weight, feed efficiency, adiposity, as well as fasted blood levels of glucose, cholesterol, and triglycerides.

Day of the week (time) had a significant effect on daily food intake [RM ANOVA Time: $F_{(6,132)} = 22.9$, $p < 0.001$, $\eta p2 = 0.510$]. Importantly, the effect of time on daily food intake depended on treatment [RM ANOVA Time * Treatment: $F(6,132) = 42.2$, $p < 0.001$, $\eta p2 = 0.657$]. Daily mice showed little to no variation in food intake across days of the week (Fig 4A). In contrast, weekly mice ate more on Monday, when food was replaced, compared to all other days of the week [Tukey Post Hoc Comparisons, all $p < 0.001$] (Fig 4A). Weekly mice also ate less after cage change (Friday) compared to the day before [Paired Samples T-test: t

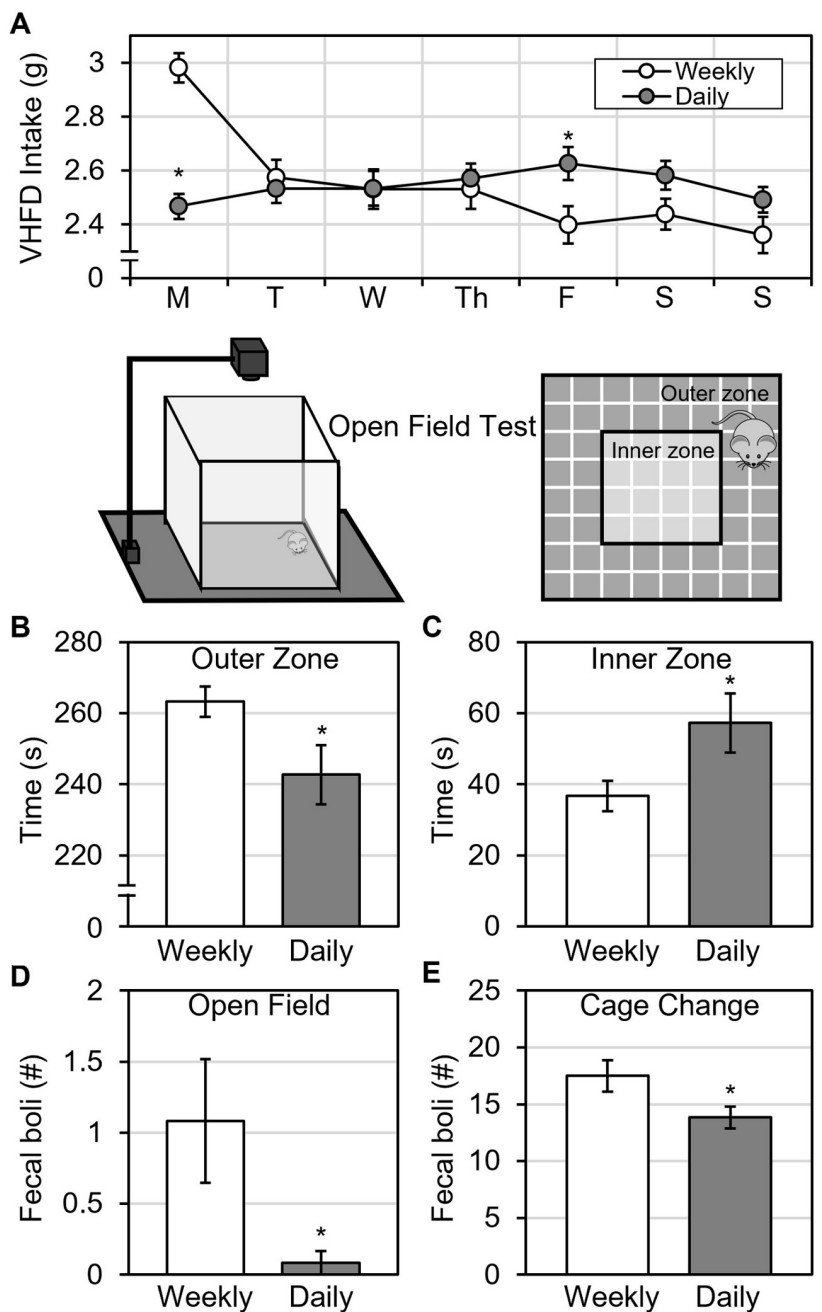

**Fig 4. Daily VHFD replacement stabilizes intake and reduces anxiety-like behaviors. A)** Only mice receiving weekly food replacement for 8-weeks exhibited hyperphagia following VHFD replacement (Mondays) and hypophagia following cage change (Fridays). Mice with daily replacement of VHFD: **B)** spent less time in the outer zone and **C)** spent more time in the inner zone of the open field test **C)** deposited less fecal boli during the open field test, and **E)** deposited less fecal boli after a cage change. All values represent AVG ± SEM. *p < 0.05.

(11) = 3.4, p = 0.006, d = 0.982] (Fig 4A). Comparing food intake between treatment groups, weekly mice ate significantly more VHFD than daily mice on Monday [Independent Samples T-test: t(22) = 7.15, p <0.001, d = 2.92] and less on Friday [Independent Samples T-test: t(22)

= 2.48, p = 0.021, d = 1.01] (Fig 4A). These results show that whereas weekly food replacement allowed fluctuations in daily VHFD intake, daily food replacement stabilized daily VHFD intake.

Although the pattern of daily intake across days of the week was different between groups, cumulative intake after 8 weeks was unchanged. Final body weight, feed efficiency, adiposity, glucose, cholesterol, and triglycerides were also not different between groups (Table 1). These results were admittedly surprising since we predicted that providing mice with highly palatable VHFD every day might have increased intake and accelerated the development of diet-induced disease. In the end, we were also surprised to see that there were no between group differences in cumulative food intake or final bodyweight, especially since weekly but not daily mice exhibited hyperphagia each week. The decrease in food intake following cage change (on Friday) appeared to compensate for the increase following food replacement (on Monday).

## Daily food replacement reduced measures of anxiety-like behavior

The decrease in food intake following cage change (Fig 1B on Saturdays and Fig 4A on Fridays) for mice with weekly food replacement was a particularly interesting result we explored further. The procedure (cage change) and result (hypophagia) are like the Novelty Induced Hypophagia test, a measure of anxiety-like behavior [18, 19]. We therefore hypothesized that the food replacement schedule also impacted anxiety-like behavior.

To directly test our hypothesis, we video-recorded mouse exploratory behavior in the Open Field Test, a validated and widely used test of anxiety-like behavior [10]. We used time spent in the outer zone of the open field to measure thigmotaxis, an anxiety-like behavior. We also counted fecal boli deposited during the test session and after a cage change. Defecation is an established measure of stress and emotional anxiety-like behaviors in rodents [10].

Daily mice spent significantly less time in the outer zone (Fig 4B) and more time in the inner zone of the open field when compared to weekly mice [Independent Samples T-test: t (22) = 2.19, p = 0.040, d = 0.894] (Fig 4C). Daily mice also deposited significantly less fecal boli during the open field test [Welch's T-test: t(11.8) = 2.26, p = 0.044, d = 0.923] (Fig 4D) and after a cage change [Mann-Whitney U-test: U = 34.0, p = 0.029, r = 0.528] (Fig 4E). These results indicate that mice on a weekly VHFD replacement schedule were more vulnerable to anxiety-like behavior from acute stressors, including a routine husbandry procedure like cage change. Importantly though, results also clearly show that daily VHFD replacement reduced stress and anxiety. As a matter of animal welfare, daily replacement of VHFD is recommended to reduce stress and anxiety.

The benefits of daily food replacement may not be limited to studies using VHFD. Future studies should investigate whether additional diet types are also vulnerable to dehydration and

**Table 1. Summary of dependent variables measured after 8-weeks of weekly versus daily VHFD replacement.**

|  | VHFD Replacement Schedule | |  |
|  | Weekly | Daily | Statistics |
| --- | --- | --- | --- |
| Total Food Intake (g) | 144.9 ± 3.3 | 145.3 ± 2.9 | Independent Samples T-test [t(22) = 0.0926, p = 0.927, d = 0.0378] |
| Final Body Weight (g) | 40.0 ± 1.3 | 39.3 ± 1.5 | Independent samples T-test [t(22) = 0.369, p = 0.715, d = 0.151] |
| Feed Efficiency (mg/g) | 119.3 ± 5.9 | 113.5 ± 6.9 | Mann-Whitney U-test [U = 58.0, p = 0.443, r = 0.194] |
| Total Adiposity (g) | 5.5 ± 0.3 | 5.4 ± 0.3 | Mann-Whitney U-test [U = 70.0, p = 0.931, r = 0.0278] |
| Blood Glucose (mg/dL) | 146.8 ± 6.3 | 135.8 ± 6.3 | Mann-Whitney U-test [U = 41.5, p = 0.166, r = 0.340] |
| Serum Cholesterol (mg/dL) | 209.3 ± 17.7 | 211.5 ± 9.5 | Independent Samples T-test [t(22) = 0.144, p = 0.887, d = 0.0587] |
| Serum Triglycerides (mg/dL) | 80.3 ± 3.7 | 80.6 ± 3.2 | Independent Samples T-test [t(22) = 0.0686, p = 0.986, d = 0.0280] |

oxidation. The effects of dehydration and oxidation on palatability, feeding behavior and anxiety may occur in other diet types too. Cage type is another variable for future consideration. Mice in this study were housed within static, filter-top cages. The microclimate within individually ventilated cages, however, may impact food quality and mouse behavior in different ways. Finally, the current study was performed using male mice. Anxiety and feeding behaviors in mice show sexual dimorphism [20]. The impacts of food quality on the behavior of female mice are unknown and warrant future investigation.

## Conclusions

Mice exhibit variations in food intake and food preference because of changes in the quality of VHFD over time. Dehydration and oxidation are critical factors influencing the quality of VHFD. Mitigating dehydration and oxidation by replacing VHFD daily, rather than weekly, stabilizes feeding behavior. Thus, our results establish daily replacement of VHFD as a best practice for ensuring VHFD quality and palatability. Importantly, daily replacement of VHFD also reduces stress and anxiety-like behavior, indicating that refining husbandry practices to include daily replacement of VHFD can also improve animal welfare. Standardizing the practice of daily VHFD replacement may also prevent experimental confound and improve experimental reproducibility and replicability.

## Supporting information

**S1 Fig. Surface temperature of VHFD over time, as it thaws to room temperature.** VHFD removed from storage at -20˚C returns to room temperature (21.7˚C; dotted line) after ∼60 minutes. All values represent AVG ± SEM.
(TIF)

**S2 Fig. Dopamine release in the nucleus accumbens after consuming VHFD.** The 3-MT / DA ratio, indicative of dopamine release, was significantly higher in the nucleus accumbens after consuming 7-day-old VHFD compared to freshly thawed VHFD. [Independent samples T-test: $t(6) = -3.98$, $p = 0.007$, $d = -2.81$] All values represent AVG ± SEM.
(TIF)

**S3 Fig. Effect of relative humidity on the mass of VHFD over time.** VHFD stored at a different relative humidity (RH) gained or lost a different amount of mass over 7-days. [ANOVA F $(5,30) = 183$, $p < 0.001$, $ηp2 = 0.968$] All values represent AVG ± SEM.
(TIF)

**S1 Table. Very high-fat diet formulation.** Adapted from: https://researchdiets.com/formulas/d12492.
(TIF)

## Acknowledgments

The author would like to thank Dr. Forest Robertson, Emily Hoegler, Cecilia Schoeni, and Derek Gustafson for excellent technical support.

## Author Contributions

**Conceptualization:** Joshua Cordeira.

**Data curation:** Joshua Cordeira.

**Formal analysis:** Joshua Cordeira.

**Funding acquisition:** Joshua Cordeira.

**Investigation:** Joshua Cordeira.

**Methodology:** Joshua Cordeira.

**Project administration:** Joshua Cordeira.

**Supervision:** Joshua Cordeira.

**Visualization:** Joshua Cordeira.

**Writing – original draft:** Joshua Cordeira.

**Writing – review & editing:** Joshua Cordeira.

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
