## [Decision Letter · Decision Letter 0]

14 Aug 2023

PONE-D-23-21692Daily replacement of very high-fat diet stabilizes food intake and improves mouse welfare by ensuring food qualityPLOS ONE

Dear Dr. Cordeira,

Thank you for submitting your manuscript to PLOS ONE. After careful consideration, we feel that it has merit but does not fully meet PLOS ONE’s publication criteria as it currently stands. Therefore, we invite you to submit a revised version of the manuscript that addresses the points raised during the review process.

We look forward to receiving your revised manuscript.

Kind regards,

Mohammed Fouad El Basuini, Professor

Academic Editor

PLOS ONE

Journal Requirements:

3. We note that you have referenced (unpublished) on page 11 which has currently not yet been accepted for publication. Please remove this from your References and amend this to state in the body of your manuscript: (ie “Bewick et al. [Unpublished]”) as detailed online in our guide for authors

http://journals.plos.org/plosone/s/submission-guidelines#loc-reference-style.

Additional Editor Comments:

Thanks for the good work. Please address the comments of the reviewers.

Reviewers' comments:

Reviewer's Responses to Questions

**Comments to the Author**

1. Is the manuscript technically sound, and do the data support the conclusions?

Reviewer #1: Yes

Reviewer #2: Yes

2. Has the statistical analysis been performed appropriately and rigorously? 

Reviewer #1: Yes

Reviewer #2: Yes

3. Have the authors made all data underlying the findings in their manuscript fully available?

Reviewer #1: Yes

Reviewer #2: Yes

4. Is the manuscript presented in an intelligible fashion and written in standard English?

Reviewer #1: Yes

Reviewer #2: Yes

5. Review Comments to the Author

Reviewer #1: The present study offers new and sound insights on the laboratory animal nutrition and welfare which will have a great impact on further upcoming studies. In my opinion graphical illustrations will make the manuscript more attractive to readership.

Reviewer #2: Dear Authors

Manuscript Number: PONE-D-23-21692 “Daily replacement of very high-fat diet stabilizes food intake and improves mouse welfare by ensuring food quality.

-The authors conducted experiments to establish best practices for ensuring the quality of VHFD and to improve mouse welfare. They found that VHFD in the food hopper was vulnerable first to dehydration and then oxidation within 7-days, leading to dramatic changes in food intake and food preference behavior in mice.

-This present study has not novel in its recommendations.

-A separate part for the experiment design should be added to be clearer the manuscript.

-This present study has not novel and the behavior study is not clear as to how the authors did it. The authors can use camera video to study the behaviors in mice.

-The statistical tests used should be reported in the statistical analysis part.

6. PLOS authors have the option to publish the peer review history of their article (what does this mean?). If published, this will include your full peer review and any attached files.

Reviewer #1: No

Reviewer #2: No

---

## [Author Response · Author response to Decision Letter 0]

18 Aug 2023

Response to Reviewers

I have ensured that the manuscript meets PLOS ONE’s style requirements. 

No changes to the Data Availability statement are required. The DOI to access data (http://doi.org/10.6084/m9.figshare.23662566) is already included.

3. We note that you have referenced (unpublished) on page 11 which has currently not yet been accepted for publication. Please remove this from your References and amend this to state in the body of your manuscript: (ie “Bewick et al. [Unpublished]”) as detailed online in our guide for authors.

“Cordeira et al. [Unpublished]” has been added to the body of the manuscript on page 11. The citation is excluded from References. 

Reviewer comments: 

Reviewer #1: 

The present study offers new and sound insights on the laboratory animal nutrition and welfare which will have a great impact on further upcoming studies. In my opinion graphical illustrations will make the manuscript more attractive to readership.

Thank you for the wonderful suggestion! We revised Fig1, Fig2, Fig4, and S2Fig to include illustrations which we hope will make the manuscript more attractive. 

Reviewer #2: 

-This present study has not novel in its recommendations.

We are not the first to caution researchers about ensuring the quality of high-fat diets. However, methods used by researchers to ensure the quality of high-fat diets are underreported, variable, and unvalidated. Our study is the first to document pitfalls and welfare concerns which can specifically be avoided by daily replacement of a very high-fat diet. 

-A separate part for the experiment design should be added to be clearer the manuscript.

Thank you for this suggestion. A new section entitled “Experimental design” has been added to the Methods to improve clarity. 

-This present study has not novel and the behavior study is not clear as to how the authors did it. The authors can use camera video to study the behaviors in mice.

We hope the new experimental design section clarifies our methods for studying behavior. Yes, video cameras were used to study anxiety-like behavior in the open field test. We improved our word choice in the Results section to be clearer about this. Full details are described in the Methods. 

-The statistical tests used should be reported in the statistical analysis part.

The statistical analysis section of the Methods has been revised to report and describe all statistical tests used.

---

## [Decision Letter · Decision Letter 1]

29 Aug 2023

Daily replacement of very high-fat diet stabilizes food intake and improves mouse welfare by ensuring food quality

PONE-D-23-21692R1

Dear Dr. Cordeira,

We’re pleased to inform you that your manuscript has been judged scientifically suitable for publication and will be formally accepted for publication once it meets all outstanding technical requirements.

Kind regards,

Mohammed Fouad El Basuini, Professor

Academic Editor

PLOS ONE

Additional Editor Comments (optional):

Reviewers' comments:

Reviewer's Responses to Questions

**Comments to the Author**

1. If the authors have adequately addressed your comments raised in a previous round of review and you feel that this manuscript is now acceptable for publication, you may indicate that here to bypass the “Comments to the Author” section, enter your conflict of interest statement in the “Confidential to Editor” section, and submit your "Accept" recommendation.

Reviewer #1: All comments have been addressed

Reviewer #2: All comments have been addressed

2. Is the manuscript technically sound, and do the data support the conclusions?

Reviewer #1: Yes

Reviewer #2: Yes

3. Has the statistical analysis been performed appropriately and rigorously? 

Reviewer #1: Yes

Reviewer #2: Yes

4. Have the authors made all data underlying the findings in their manuscript fully available?

Reviewer #1: Yes

Reviewer #2: Yes

5. Is the manuscript presented in an intelligible fashion and written in standard English?

Reviewer #1: Yes

Reviewer #2: Yes

6. Review Comments to the Author

Reviewer #1: Authors responded well to all comments by adjusting the figures and creating diagrams making it more interesting to readership

Reviewer #2: The authors responded for the comments and the manuscript is accepted in my side.

the decision letter is attached.

7. PLOS authors have the option to publish the peer review history of their article (what does this mean?). If published, this will include your full peer review and any attached files.

Reviewer #1: No

Reviewer #2: No

---

## [Editor Report · Acceptance letter]

6 Sep 2023

PONE-D-23-21692R1 

Daily replacement of very high-fat diet stabilizes food intake and improves mouse welfare by ensuring food quality 

Dear Dr. Cordeira:

I'm pleased to inform you that your manuscript has been deemed suitable for publication in PLOS ONE. Congratulations! Your manuscript is now with our production department. 

Kind regards, 

on behalf of

Dr Mohammed Fouad El Basuini 

Academic Editor

PLOS ONE